# Gastrointestinal Manifestations Are Associated with Severe COVID-19 in Children

**DOI:** 10.3390/healthcare12010081

**Published:** 2023-12-29

**Authors:** Esra Betul Akkoyun, Bilal Ashraf, Natasha Hanners, Jeffrey Kahn, Zachary Most

**Affiliations:** 1Division of Pediatric Infectious Disease, Department of Pediatrics, University of Texas Southwestern Medical Center, Dallas, TX 75390, USA; natasha.hanners@utsouthwestern.edu (N.H.); jeffrey.kahn@utsouthwestern.edu (J.K.); zachary.most@utsouthwestern.edu (Z.M.); 2Department of Internal Medicine/Pediatrics, University of Texas Southwestern Medical Center, Dallas, TX 75390, USA

**Keywords:** children, COVID-19, severe, gastrointestinal symptoms

## Abstract

**Purpose:** Although less severe than in adults, children can experience a range of COVID-19 symptoms, from asymptomatic to life-threatening, including respiratory and gastrointestinal symptoms. Medical conditions may also increase the severity of the disease in infected children. **Methods:** This study was performed at a single center, comparing cases and controls, and involving 253 pediatric patients who had been diagnosed with COVID-19. Two different outcomes were assessed. The first categorized symptomatic individuals who were hospitalized with COVID-19 (hospital) from those who were not (nonhospital). The second categorized admitted individuals who spent at least one day in the intensive care unit (ICU) from those who did not require intensive care (floor). **Results:** Ninety individuals (36%) had at least one underlying medical condition, the most common being pulmonary disorders, such as asthma (12%), followed by neurodevelopmental disorders (8%), gastrointestinal disorders (6%), and seizure disorders (6%). The hospital group was more likely to have a comorbidity, such as obstructive sleep apnea (OSA), diabetes mellitus, seizure disorder, hypertension, sickle cell disease, neurodevelopmental disorder, and immunocompromising conditions, including cancer, bone marrow transplant, and other immunodeficiencies, compared to the non-hospital group. Abdominal pain was more common in the hospital group. Shortness of breath (SOB) and diarrhea were significantly more common in the ICU group than in the floor group. **Conclusions:** Early identification of pediatric patients with severe COVID-19 is important to improve outcomes. In our single-center case–control study, we found that the presence of gastrointestinal symptoms on presentation was more commonly associated with severe COVID-19 in children.

## 1. Introduction

More than 15 million cases of severe acute respiratory syndrome coronavirus 2 (SARS-CoV-2) infection have been reported in children in the United States as of October 2022 [1]. Although these coronavirus disease 2019 (COVID-19) cases tend to be less severe than those in adults [2], they can range from asymptomatic to life-threatening infections [3,4]. Infected children can have clinical manifestations of upper or lower respiratory infections, as well as gastrointestinal symptoms such as abdominal pain, vomiting, and diarrhea [5,6]. Since severe infections are rare in children, early identification of children who may progress to critical COVID-19 is crucial. Data from a multicenter study conducted in Italy revealed that approximately 15% of children diagnosed with COVID-19 experience diarrhea, and this symptom was more prevalent among infants. In contrast, vomiting (10%) and abdominal pain (8%) appeared to be more frequent among school-aged children [7]. Additionally, underlying medical comorbidities may be linked to more severe disease in children and adults [8,9,10]. The purpose of this study was to determine whether any symptoms at presentation or underlying medical comorbidities are associated with COVID-19 severity in children.

## 2. Materials and Methods

### 2.1. Patient Population

This single-center case–control study of pediatric patients with COVID-19 was conducted from 1 July to 14 August 2020 at a 490-bed tertiary care pediatric hospital in Dallas, Texas, while the original Wuhan strain was circulating. This study was part of a larger study that evaluated risk factors for COVID-19 early in the pandemic. This study was reviewed and approved by the University of Texas Southwestern Medical Center Institutional Review Board with a partial waiver of informed consent.

### 2.2. Inclusion Criteria

Inclusion criteria included a positive SARS-CoV-2 polymerase chain reaction (PCR) test from a nasopharyngeal specimen taken at any clinical site (inpatient, outpatient, or emergency department), age 19 years old or younger, and at least one symptom of COVID-19 (fever, cough, shortness of breath, sore throat, rhinorrhea, anosmia, dysgeusia, diarrhea, nausea, vomiting, abdominal pain, rash, headache, or myalgias). No individuals had received COVID-19 vaccinations, and patients with multisystem inflammatory syndrome (MIS-C) were excluded.

### 2.3. Data Collection

Demographic, clinical, and comorbidity data were abstracted from the electronic health record (EHR) and securely entered into REDCap electronic data capture tools [11]. Demographic information included age, sex, body mass index, race, and ethnicity. Clinical data comprised the highest level of care, signs and symptoms at presentation (e.g., fever, cough, shortness of breath, sore throat, rhinorrhea, anosmia, dysgeusia, diarrhea, nausea, vomiting, abdominal pain, rash, headache, and myalgias), and the presence of pre-existing comorbidities (such as preterm birth, heart disease, hypertension, pulmonary disease, kidney disease, liver disease, gastrointestinal disease, neurological disorder, hematological disease, diabetes, immune deficiency, malignancy, and post-transplant status).

### 2.4. Outcomes

Two different severity outcomes were assessed. The first category compared individuals who were hospitalized with COVID-19 (hospital) with those who were not (nonhospital). The second categorized individuals within the hospital group who spent at least one day in the intensive care unit (ICU) and those who did not require intensive care (floor).

### 2.5. Statistical Analysis

Descriptive statistics were presented as medians and interquartile ranges (IQRs), and categorical variables were presented as counts and percentages. Univariate odds ratios (ORs) and 95% confidence intervals (CIs) were calculated. The Mann–Whitney U Test was employed for continuous variables and chi-square or Fisher’s exact test were applied for categorical variables. A *p* value of <0.05 was used to denote statistical significance and no correction for multiple comparisons was conducted, as this was an exploratory analysis. Multivariable analysis was not performed due to the limited sample size. This case–control study was nested in a larger case–control study that had an independent power calculation, and all available cases and controls were used. Calculations were conducted via Stata version 16.1 (StataCorp, College Station, TX, USA).

## 3. Results

Of the 253 patients included, the median age was 6.8 years and 52% were female. Most patients were of Hispanic ethnicity (70%) (Table 1). Ninety individuals (36%) had at least one underlying medical condition, the most common being pulmonary disorders, such as asthma (12%), followed by neurodevelopmental disorders (8%), gastrointestinal disorders (6%), seizure disorders (6%), and preterm birth (5%) (Table 1). The most common symptom reported was fever (68%), followed by cough (39%), nausea/vomiting (28%), headache (18%), rhinorrhea (18%), sore throat (17%), abdominal pain (16%), and shortness of breath (SOB) (14%) (Table 1). In addition, 27% of the patients (69 individuals) were hospitalized, with 15 (22%) of these requiring ICU care (Table 2).

### 3.1. Hospital vs. Non-Hospital Groups

Table 1 summarizes demographic characteristics, underlying medical conditions, and symptoms at presentation in the hospital and non-hospital groups. The hospital group trended towards being older compared to the non-hospital group (9.5 years vs. 6.1 years, *p* = 0.11). There were no sex differences between the two groups. There were also no statistical differences in race/ethnicity.

The hospital group was more likely to have any comorbidity (OR 3.48, 95% CI 1.96–6.20) and specific comorbidities including obstructive sleep apnea (OSA), diabetes mellitus, seizure disorder, hypertension, sickle cell disease, neurodevelopmental disorder, and immunocompromising conditions including cancer, bone marrow transplant, and other immunodeficiencies. Furthermore, abdominal pain was more common in the hospital group (OR 2.18, 95% CI 1.09–4.37), while fever, rhinorrhea, sore throat, and cough were more common in the non-hospital group (Table 1).

### 3.2. ICU vs. Floor Groups

Table 2 presents a summary of demographic characteristics, underlying medical conditions, and symptoms at presentation in the ICU and floor groups. There were no significant differences in age, sex, race/ethnicity, and BMI between the two groups. There were no differences in diabetes mellitus, hypertension, kidney disease, or immunocompromised status between the two groups. However, the ICU group was more likely to have seizure disorders, OSA, gastrointestinal disorders, and neurological disability than the floor group. Additionally, there were no significant differences in fever, rash, cough, sore throat, myalgia, headache, abdominal pain, nausea, and diarrhea between the ICU and floor group. However, shortness of breath (SOB) (OR 19.43, 95% CI 4.15–91.01) and diarrhea (OR 11.33, 95% CI 2.39–53.75) were significantly more common in the ICU group than in the floor group.

## 4. Discussion

In this case–control study, risk factors associated with severe COVID-19 in children were identified during the first wave of infections in summer 2020. Symptoms associated with hospital admission included the presence of abdominal pain and a lack of sore throat or rhinorrhea. Furthermore, symptoms associated with ICU stay included shortness of breath and diarrhea. Additionally, medical comorbidities associated with hospital admission or ICU stay included hypertension, sickle cell disease, or seizure disorders, and OSA or seizure disorders, respectively.

This study adds to the growing body of evidence indicating that gastrointestinal symptoms are a manifestation of severe COVID-19 in children [6]. This is in agreement with previous studies that have found gastrointestinal symptoms at the onset of the disease to be associated with an increased likelihood of requiring admission to the intensive care unit, and higher mortality rates [6,12].

Our findings suggest that children with certain comorbidities and clinical manifestations are at risk for adverse, potentially life-threatening consequences of COVID-19 infection. Previous studies have had mixed results regarding comorbidities that are risk factors for severe COVID-19 in children. Some studies [8,13,14,15] have indicated that certain underlying medical conditions were associated with increased risk for hospitalization or admission to the ICU in children and adolescents over the course of the first year of the pandemic. A systematic review and meta-analysis study showed that childhood obesity likely contributes to severe COVID-19 infection [16]. Our results corroborate previous studies showing that neurodevelopmental disorders are prominent among adolescents hospitalized with COVID-19 and represent a risk factor for hospitalization and progression to ICU admission [17]. Conversely, the presence of a comorbid illness was not associated with an increase in the severity of COVID illness, length of hospital stay, or adverse outcome in a cohort children in India [18]. This nuanced perspective underscores the need for further research to fully comprehend the complex interplay of comorbidities in the context of pediatric COVID-19.

Awareness that gastrointestinal symptoms at initial presentation, as well as underlying medical conditions, may be associated with severe COVID-19 in children can aid clinicians in quickly identifying those at risk of disease progression. Early detection of children with serious illness or the potential for it to worsen can ensure they receive timely supportive care and antiviral therapy, reducing the likelihood of complications and increasing the chances of a successful recovery.

Our data included pediatric patients infected with the original Wuhan strain but did not encompass children and adolescents infected with the Delta (B.1.617.2) or Omicron (B.1.1.529) variants. Studying the Wuhan strain provides a baseline understanding of the COVID-19 virus, serving as a reference point for comparative analyses with subsequent variants. The original strain also played a pivotal role in shaping global public health strategies, and research on its epidemiological dynamics offers insights into transmission patterns and vulnerable populations. It was noted that later variants were associated with a greater incidence of severe infections in children [19]. Each variant may be linked to different clinical manifestations and factors that could increase the risk of developing severe illness. Thus, further research is needed to investigate the risk factors associated with COVID-19 in relation to novel variants.

Our study has several limitations that may have impacted our results. First, as a single-center study in the USA in 2020, the results may not be generalizable to other regions or countries. Secondly, we were unable to differentiate between admissions due to COVID-19 and those with COVID-19 since all hospitalized patients were screened for SARS-CoV-2 infection. Furthermore, we used admission to ICU as an indicator for disease severity and were unable to gauge the level of intensive support needed while in critical care. Additionally, changes to thresholds for hospitalization and ICU admission over time and different thresholds for hospitalization or ICU admission for those with underlying comorbidities might have biased our findings. We also could not collect body mass index (BMI) data, an important risk factor for severe disease, in non-hospitalized individuals, and thus could not evaluate BMI as a risk factor in our population. There is also the potential for misclassification of underlying medical conditions from EHR abstraction, which may have caused an underdiagnosis of comorbidities in non-hospitalized patients. Finally, due to the small sample size, especially for ICU risk factors, we were unable to adjust the estimates for confounding factors.

## 5. Conclusions

In addition to respiratory symptoms, gastrointestinal symptoms were associated with severity of illness and the need for ICU care in children with COVID-19. Those who were admitted to the ICU with symptomatic COVID-19 infection were more likely to have pre-existing medical conditions such as neurodevelopmental disorders, seizure disorder, gastrointestinal diseases, and OSA. The presence of such symptoms could alert medical practitioners to the possibility of severe COVID-19 infection and prompt further evaluation.

## Figures and Tables

**Table 1 healthcare-12-00081-t001:** Demographics and clinical characteristics in non-hospital vs. hospital groups.

Demographics	All (n = 253)	Nonhospital (n = 184)	Hospital (n = 69)	Crude OR (95% CI)	*p* Value
Age (year), median (IQR)	6.8 (0.8–13.6)	6.1 (0.8–13.5)	9.5 (1.8–13.7)		0.11
Female, n (%)	132 (52)	95 (52)	37 (54)	1.08 (0.62–1.89)	0.78
Race/Ethnicity, n (%)					
Black—non-Hispanic	39 (15)	23 (13)	16 (23)	1.79 (0.61–5.27)	0.19
White—non-Hispanic	25 (10)	18 (10)	7 (10)	Ref	
Other—non-Hispanic	11 (4)	9 (5)	2 (3)	0.57 (0.10–3.33)	
Hispanic	178 (70)	134 (73)	44 (64)	0.84 (0.33–2.16)	
**Underlying medical conditions, n (%)**					
Any	89 (35)	50 (27)	39 (57)	3.48 (1.96–6.20)	<0.001
Preterm birth	12 (5)	9 (5)	3 (4)	0.88 (0.23–3.36)	0.86
Pulmonary	45 (18)	28 (15)	17 (25)	1.82 (0.92–3.59)	0.08
Asthma	31 (12)	22 (12)	9 (13)	1.10 (0.48–2.53)	0.81
OSA	10 (4)	3 (2)	7 (10)	6.81 (1.71–27.16)	0.007
Cardiac	9 (4)	6 (3)	3 (4)	1.34 (0.33–5.55)	0.68
Congenital heart disease	4 (1)	2 (1)	2 (3)	2.72 (0.38–19.67)	0.32
Hypertension	7 (3)	1 (1)	6 (9)	17.4 (2.06–147.6)	0.009
Renal	10 (4)	5 (3)	5 (7)	2.80 (0.78–9.98)	0.11
Chronic kidney disease	4 (1)	2 (1)	2 (3)	2.72 (0.38–19.67)	0.32
Gastrointestinal	14 (6)	6 (3)	8 (12)	3.89 (1.30–11.66)	0.02
Hematologic	9 (3)	4 (2)	4 (6)	2.77 (0.67–11.40)	0.16
Sickle cell disease	5 (2)	1 (1)	4 (6)	11.3 (1.24–102.6)	0.03
Seizure disorder	17 (6)	5 (3)	9 (13)	5.37 (1.73–16.65)	0.004
Neurodevelopmental disorders	19 (8)	10 (5)	9 (13)	2.61 (1.01–6.73)	0.05
Diabetes mellitus	5 (2)	0	5 (7)	n/a	n/a
Immunocompromised					
Cancer	6 (2)	2 (1)	4 (6)	5.69 (1.02–31.80)	0.05
BMT	2 (1)	0	2 (3)	n/a	n/a
Other immunodeficiency	8 (3)	3 (2)	5 (7)	4.71 (1.10–20.29)	0.04
SOT	3 (1)	1 (1)	2 (3)	5.46 (0.49–61.23)	0.17
Symptoms at presentation, n (%)					
Fever	171 (68)	131 (80)	40 (58)	0.56 (0.31–0.99)	0.05
Anosmia	15 (6)	11 (6)	4 (6)	0.97 (0.30–3.15)	0.96
Dysgeusia	12 (5)	8 (4)	4 (6)	1.35 (0.39–4.65)	0.63
Rash	14 (6)	10 (5)	4 (6)	1.07 (0.32–3.53)	0.91
Cough	99 (39)	79 (48)	20 (29)	0.54 (0.30–0.98)	0.04
SOB	35 (14)	24 (13)	11 (16)	1.26 (0.58–2.74)	0.55
Sore throat	44 (17)	39 (21)	5 (7)	0.29 (0.11–0.77)	0.01
Rhinorrhea	46 (18)	41 (22)	5 (7)	0.27 (0.10–0.72)	0.009
Headache	45 (18)	36 (20)	9 (13)	0.62 (0.28–1.36)	0.23
Myalgia	17 (7)	12 (7)	5 (7)	1.12 (0.38–3.30)	0.84
Diarrhea	36 (14)	27 (15)	9 (13)	0.87 (0.39–1.96)	0.74
Nausea/vomiting	71 (28)	46 (25)	25 (36)	1.70 (0.94–3.09)	0.08
Abdominal pain	41 (16)	24 (13)	17 (24)	2.18 (1.09–4.37)	0.03

OSA indicates obstructive sleep apnea; BMT, bone marrow transplant; SOT, solid organ transplant; SOB, shortness of breath.

**Table 2 healthcare-12-00081-t002:** Demographics and clinical characteristics in floor and ICU groups.

Demographics	Floor (n = 54)	ICU (n = 15)	Crude OR (95% CI)	*p* Value
Age (year), median (IQR)	9 (0–14)	9 (5.5–12.5)		0.69
Female, n (%)	31 (57)	6 (40)	0.49 (0.15–1.59)	0.24
Race/Ethnicity, n (%)				
Black—non-Hispanic	12 (22)	4 (27)	0.44 (0.07–2.90)	0.17
White—non-Hispanic	4 (7)	7 (10)	Ref	
Other—non-Hispanic	1 (2)	2 (3)	1.33 (0.57–31.1)	
Hispanic	37 (69)	44 (64)	0.25 (0.05–1.38)	
BMI, median (IQR)	20.4 (16.9–23.8)	19.9 (15.8–21.4)		0.34
**Underlying medical conditions, n (%)**				
Any	28 (52)	11 (73)	2.55 (0.72–9.03)	0.15
Preterm birth	2 (4)	1 (7)	1.86 (0.16–22.00)	0.62
Pulmonary	10 (19)	7 (47)	3.85 (1.13–13.10)	0.03
Asthma	6 (11)	3 (20)	2.00 (0.44–9.18)	0.37
OSA	2 (4)	5 (33)	13.00 (2.21–76.63)	<0.001
Cardiac	3 (6)	0	n/a	n/a
Congenital heart disease	2 (4)	0	n/a	n/a
Hypertension	5 (9)	1 (7)	0.70 (0.08–6.49)	0.75
Renal	4 (8)	1 (7)	0.89 (0.09–8.64)	0.92
Chronic kidney disease	2 (4)	0	n/a	n/a
Gastrointestinal	3 (6)	5 (33)	8.50 (1.74–41.42)	0.008
G tube dependence	0	4 (27)	n/a	n/a
Sickle cell disease	2 (4)	2 (14)	4.00 (0.51–31.13)	0.19
Neurologic				
Seizure disorder	3 (6)	6 (40)	11.33 (2.39–53.75)	0.002
Neurodevelopmental disorders	4 (8)	5 (33)	6.25 (1.42–27.45)	<0.001
Diabetes mellitus	4 (7)	1 (7)	0.89 (0.09–8.64)	0.92
Immunocompromised				
Cancer	3 (6)	1 (7)	1.19 (0.11–12.35)	0.88
BMT	0	2 (14)	n/a	n/a
Other immunodeficiency	4 (7)	1 (7)	0.89 (0.09–8.64)	0.92
SOT	2 (4)	0	n/a	n/a
**Symptoms at presentation, n (%)**				
Fever	31 (57)	9 (60)	1.11 (0.35–3.57)	0.86
Anosmia	4 (7)	0	n/a	n/a
Dysgeusia	4 (7)	0	n/a	n/a
Rash	3 (6)	1 (7)	1.21 (0.12–12.60)	0.87
Cough	13 (24)	7 (47)	2.76 (0.84–9.08)	0.10
SOB	3 (6)	8 (53)	19.43 (4.15–91.01)	<0.001
Sore throat	4 (7)	1 (7)	0.89 (0.09–8.64)	0.92
Rhinorrhea	4 (7)	1 (7)	0.89 (0.09–8.64)	0.92
Headache	6 (11)	3 (20)	2.00 (0.44–9.18)	0.37
Myalgia	3 (6)	2 (13)	2.62 (0.40–17.31)	0.32
Diarrhea	3 (6)	6 (40)	11.33 (2.39–53.75)	0.002
Nausea/vomiting	19 (35)	6 (40)	1.23 (0.38–3.97)	0.73
Abdominal pain	14 (25)	3 (20)	0.71 (0.18–2.91)	0.64

OSA indicates obstructive sleep apnea; BMT, bone marrow transplant; SOT, solid organ transplant; SOB, shortness of breath; ICU, intensive care unit.

## Data Availability

The datasets generated and/or analyzed during the current study are available in the Zenodo repository, https://zenodo.org/record/7867902#.ZEksV-zMI-Q, (accessed on 25 April 2023).

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
