# Peer review of "Gastrointestinal Manifestations Are Associated with Severe COVID-19 in Children"

_healthcare, 2023, doi:10.3390/healthcare12010081_

Round 1

Reviewer 1 Report

Comments and Suggestions for Authors

Dear authors

Thank you for the opportunity to get acquainted with your interesting work.whether there Were children vaccinated against Covid 19 in the study group and whether there were children who had PIMS, if so, they should be analyzed separately. The limitations of the work should be mentioned in the discussion

Author Response

Response to Reviewers’ Comments

Dear Editorial Board members and Reviewers for Healthcare. Thank you for review and comments on our manuscript. We are pleased our brief research article was reviewed favorably. Below we provide responses to the reviews in an itemized manner: 

 Reviewer 1

Thank you for the opportunity to get acquainted with your interesting work. whether there Were children vaccinated against Covid 19 in the study group and whether there were children who had PIMS, if so, they should be analyzed separately. The limitations of the work should be mentioned in the discussion.

Response: There were no patients vaccinated in the study group. None of the patients were diagnosed with MIS-C/PIMS. These statements were added to the manuscript.

Reviewer 2 Report

Comments and Suggestions for Authors

Dear Editors,

Thank you very much for allowing me to review this article. It was interesting as there are fewer studies related to COVID-19 in pediatric patients, which is equally essential as adult studies. I compliment the authors' ability to synthesize the information effectively, resulting in a smooth read that can reach a wider audience. However, some minor aspects can be improved to enhance the overall quality of the article.

In the introduction, it would be helpful to provide more details on the rationale behind this work. While the significance and relevance of the study are easy to comprehend, it would be helpful if the authors could spend a few lines explaining the importance and the necessity of this research.

It is advisable to use the impersonal form instead of the personal form for describing methods and results. This approach is more objective and neutral. Therefore, I suggest the authors avoid using the personal form ("We conducted a study...") and opt for the impersonal form when describing such sections.

Since selection bias is unavoidable, In the methods section, it would be beneficial to describe the primary characteristics that define the chosen center and the eligible population covered by it. This revision would help objectively evaluate potential selection bias and external validity of the study's conclusions.

In research methods, it is essential to consider the key ethical aspects.

I would like to ask the authors to provide a table, if possible, containing more demographic details of their sample in the results section. These details will not only help to give context to the sample but also to evaluate any potential selection bias and the external validity of the conclusions drawn from the study.

In the limitations section, it would be helpful to mention and evaluate the potential selection bias previously discussed. This assessment could be achieved by analyzing the characteristics of the center and eligible population, as well as the primary demographic qualities of the sample chosen for the study. Such an evaluation would give the reader a better understanding of the study, particularly in comparison to other similar studies mentioned in the discussion.

Author Response

Response to Reviewers’ Comments

Dear Editorial Board members and Reviewers for Healthcare. Thank you for review and comments on our manuscript. We are pleased our brief research article was reviewed favorably. Below we provide responses to the reviews in an itemized manner: 

Reviewer 2

Dear Editors,

Thank you very much for allowing me to review this article. It was interesting as there are fewer studies related to COVID-19 in pediatric patients, which is equally essential as adult studies. I compliment the authors' ability to synthesize the information effectively, resulting in a smooth read that can reach a wider audience. However, some minor aspects can be improved to enhance the overall quality of the article.

In the introduction, it would be helpful to provide more details on the rationale behind this work. While the significance and relevance of the study are easy to comprehend, it would be helpful if the authors could spend a few lines explaining the importance and the necessity of this research.

Response: We added a statement to the introduction regarding the importance and necessity of the research.

It is advisable to use the impersonal form instead of the personal form for describing methods and results. This approach is more objective and neutral. Therefore, I suggest the authors avoid using the personal form ("We conducted a study...") and opt for the impersonal form when describing such sections.

Response: We have made these changes to the manuscript.

Since selection bias is unavoidable, In the methods section, it would be beneficial to describe the primary characteristics that define the chosen center and the eligible population covered by it. This revision would help objectively evaluate potential selection bias and external validity of the study's conclusions.

Response: We have updated the methods section with a better description of chosen center and eligible population.

In research methods, it is essential to consider the key ethical aspects.

Response: Thank you for highlighting this omission. We have added a statement to the methods confirming that our research protocol was reviewed and approved by our local Institutional Review Board.

I would like to ask the authors to provide a table, if possible, containing more demographic details of their sample in the results section. These details will not only help to give context to the sample but also to evaluate any potential selection bias and the external validity of the conclusions drawn from the study.

Response: Our data is limited to the demographic details as shown in table 1 and 2.

In the limitations section, it would be helpful to mention and evaluate the potential selection bias previously discussed. This assessment could be achieved by analyzing the characteristics of the center and eligible population, as well as the primary demographic qualities of the sample chosen for the study. Such an evaluation would give the reader a better understanding of the study, particularly in comparison to other similar studies mentioned in the discussion.

Response: We have expanded the discussion on bias and generalizability in the discussion.

Reviewer 3 Report

Comments and Suggestions for Authors

Comments for Author

Ø  Clearly defined inclusion criteria, including a positive SARS-CoV-2 PCR test and at least one symptom of COVID-19 to ensure that the study focuses on relevant cases.

Ø  Specify why the study was conducted during the specified time frame (July 1 to August 14, 2020) and how the choice of this period aligns with the research objectives. Additionally, consider providing a brief rationale for focusing on the original Wuhan strain.

Ø  Ensure that the methodology section is organized in a logical flow. Consider separating information related to data collection, inclusion criteria, and statistical analysis for better readability.

Ø  Maintain consistency in the presentation of results. For instance, in Table 1, the crude OR is presented for certain variables, but this is not continued consistently for all variables.

Author Response

Response to Reviewers’ Comments

Dear Editorial Board members and Reviewers for Healthcare. Thank you for review and comments on our manuscript. We are pleased our brief research article was reviewed favorably. Below we provide responses to the reviews in an itemized manner: 

Reviewer 3

Ø  Clearly defined inclusion criteria, including a positive SARS-CoV-2 PCR test and at least one symptom of COVID-19 to ensure that the study focuses on relevant cases.

Response: Inclusion criteria was defined as a positive SARS-CoV-2 polymerase chain reaction (PCR) test from a nasopharyngeal specimen taken at any clinical site (inpatient, outpatient, or emergency department), age 19 years-old or younger, and at least one symptom of COVID-19. Symptoms including fever, cough, shortness of breath, sore throat, rhinorrhea, anosmia, dysgeusia, diarrhea, nausea, vomiting, abdominal pain, rash, headache, and myalgias.

Ø  Specify why the study was conducted during the specified time frame (July 1 to August 14, 2020) and how the choice of this period aligns with the research objectives. Additionally, consider providing a brief rationale for focusing on the original Wuhan strain.

Response: The study was conducted in the time frame (July 1 to August 14, 2020) which is before Delta and Omicron variants. The reason this period was used was because this study was part of a larger study conducted early in the pandemic to determine risk factors for COVID-19. That study was only active during the first wave of the pandemic. Additionally. it has been noted that these later variants were associated with a greater incidence of severe infections in children. Each variant may be linked to different clinical manifestations and factors that could increase the risk of developing severe illness. A brief rationale for focusing on the original Wuhan strain was provided in the manuscript.

Ø  Ensure that the methodology section is organized in a logical flow. Consider separating information related to data collection, inclusion criteria, and statistical analysis for better readability.

Response: Methods were separated as patient population, inclusion criteria, data collection, outcomes and statistical analysis.

Ø  Maintain consistency in the presentation of results. For instance, in Table 1, the crude OR is presented for certain variables, but this is not continued consistently for all variables.

Response: In the table we present the crude odds ratio for all categorical variables that did not have 0 in the numerator or denominator.

Round 2

Reviewer 1 Report

Comments and Suggestions for Authors

I accept in present form